# Telerehabilitation as a Therapeutic Exercise Tool versus Face-to-Face Physiotherapy: A Systematic Review

**DOI:** 10.3390/ijerph20054358

**Published:** 2023-02-28

**Authors:** Mª Teresa Muñoz-Tomás, Mario Burillo-Lafuente, Araceli Vicente-Parra, Mª Concepción Sanz-Rubio, Carmen Suarez-Serrano, Yolanda Marcén-Román, Mª Ángeles Franco-Sierra

**Affiliations:** 1Physiotherapy Primary Care, 44002 Teruel, Spain; 2Physiotherapy Hospital San José, 44002 Teruel, Spain; 3Physiotherapy Hospital Obispo Polanco, 44002 Teruel, Spain; 4Physiotherapy Primary Care, Department of Nursing, Physiotherapy and Occupational Therapy, University of Zaragoza, 50009 Zaragoza, Spain; 5Department of Physiotherapy, University of Seville, 41009 Seville, Spain; 6Department of Anatomy and Human Embryology, IIS Aragón, University of Zaragoza, 50009 Zaragoza, Spain; 7Department of Nursing, Physiotherapy and Occupational Therapy, IIS Aragón, University of Zaragoza, 50009 Zaragoza, Spain

**Keywords:** telerehabilitation, exercise therapy, physiotherapy

## Abstract

Digital physiotherapy, often referred to as “Telerehabilitation”, consists of applying rehabilitation using telecommunication technologies. The objective is to evaluate the effectiveness of therapeutic exercise when it is telematically prescribed. Methods: We searched PubMed, Embase, Scopus, SportDiscus and PEDro (30 December 2022). The results were obtained by entering a combination of MeSH or Emtree terms with keywords related to telerehabilitation and exercise therapy. RCTs on patients over 18 years and two groups were included, one working with therapeutic exercise through telerehabilitation and one working with conventional physiotherapy group. Results: a total of 779 works were found. However, after applying the inclusion criteria, only 11 were selected. Telerehabilitation is most frequently used to treat musculoskeletal, cardiac and neurological pathologies. The preferred telerehabilitation tools are videoconferencing systems, telemonitoring and online platforms. Exercise programs ranged from 10 to 30 min and were similar in both intervention and control groups. In all the studies, results proved to be similar for telerehabilitation and face-to-face rehabilitation in both groups when measuring functionality, quality of life and satisfaction. Conclusion: this review generally concludes that intervention through telerehabilitation programs is as feasible and efficient as conventional physiotherapy in terms of functionality level and quality of life. In addition, telerehabilitation shows high levels of patients’ satisfaction and adherence, being values equivalent to traditional rehabilitation.

## 1. Introduction

The term “telemedicine”, which was first introduced in 1993, has been gaining importance in daily clinical practice. It is defined as the provision of health services via remote telecommunications [1], and includes both interactive consultation and diagnostic services. One of the most applied fields of telemedicine in the last decade is telerehabilitation, which began to gain relevance in 2016 and consists of applying rehabilitation using telecommunication technologies [2].

In 2017, a Joint Working Group of World Physiotherapy and the International Network of Physiotherapy Regulatory Authorities (INPTRA) on Digital Physiotherapy was established. As a result, a document containing the guidelines for the practice and regulation of physiotherapy in the digital age was developed, called the Task Force [1]. The Task Force proposed the definition and purpose of digital physiotherapy, a term used to describe healthcare, support and information services that are offered remotely through digital communication devices with the aim of facilitating the effective delivery of physiotherapy services through these means improving access to care and information.

Some reviews, such as the one by Cottrell et al. [3] published in 2017, in which real-time telerehabilitation for musculoskeletal conditions was reported to improve physical function and pain, making it effective and comparable to conventional methods. Although more evidence is needed to establish care standards, studies have indicated that telerehabilitation is as efficient as face-to-face care in terms of assessment [3,4], pain management [3,5], functionality [6] and health education [7].

When focusing on patients, the use of digital tools in the health field has proved to meet their expectations and to be satisfactory, especially in teleconsultations, which are similar to face-to-face consultations [8].

Since December 2019, the emergence of the SARS-CoV-2 forced health authorities to establish recommendations to minimize face-to-face consultations and reduce the risk of virus transmission, for which telematic or remote treatment procedures were promoted [9,10].

In 2005, in the field of physiotherapy, therapeutic exercise proved to be effective in numerous neuro–musculo–skeletal pathologies, such as knee and hip osteoarthritis, subacute and chronic low back pain, cystic fibrosis, COPD, intermittent claudication, Parkinson’s disease and stroke [11]. In addition, it is also now known to be effective when treating many other chronic diseases, such as type 2 diabetes mellitus [12], schizophrenia [13], depression, anxiety, stress, obesity, dementia, multiple sclerosis, metabolic syndrome and hypertension [14]. Therapeutic physical exercise has become a commonly used tool in physiotherapy consultations. However, since there were restrictions regarding direct contact with patients during the SARS-CoV-2 pandemic, telerehabilitation strategies had to be implemented to allow patients to perform their treatment at home while being monitored by the professional telematically, through phone calls [15], recorded videos and videocalls [16].

Many questions have arisen regarding profitability of telerehabilitation models in many aspects; are the treatments equally or more effective when addressing physical condition, quality of life, etc.? Can these models be used to treat every pathology? What is patients’ level of satisfaction with these methodologies? These questions are particularly concerned with the tool used, as there are different tool integration models in telerehabilitation; for instance, some can be more or less difficult to use, others involve different patient control procedures, etc. Limitations are not related to patients only, since there are others related to the professionals who are in charge of the treatment; the equipment needed both at consultations and at the patient’s home may vary according to the telerehabilitation tool used. Sometimes, in addition, the use of equipment requires bidirectional training.

At present, since state regulations allow for face-to-face consultations, in most situations, physiotherapy treatments could be delivered in person, as before the pandemic. In fact, nowadays telerehabilitation is not as relevant, and it may have even been eliminated from current treatments. It would be interesting to observe whether telerehabilitation treatments are at least as effective as face-to-face treatments in order to be able to use them on a daily basis, and which could also be considered both as the only or as a complementary treatment.

Therefore, the main objective of this review is to determine the effectiveness of telematically prescribed therapeutic exercise, as well as to identify those pathologies for which therapeutic exercise is most frequently applied and the most frequently used digital tools.

## 2. Materials and Methods

### 2.1. Study Design

This systematic review was conducted according to the Preferred Reporting Items for Systematic Reviews and Meta-Analyses [17]. It was registered in the International Prospective Register of Systematic Reviews (PROSPERO) with registration number CRD42015020746.

### 2.2. Search Strategy

For the present review, only studies published between January 2015 and December 2022 were considered. The search was carried out on the following databases: PubMed, Embase, Scopus, SportDiscus and the Physiotherapy Evidence Database (PEDro). The PICOs framework was used to define the search strategy. The results were obtained by entering a combination of MeSH or Emtree terms with keywords related to telerehabilitation and exercise therapy on the databases. The search terms were combined with Boolean operators AND and OR. To optimize the results, the search procedure was adjusted according to the database used (Table 1).

### 2.3. Eligibility Criteria

This systematic review included studies on the effectiveness of prescribed therapeutic physical exercise telematically delivered for the intervention group and conventionally delivered for the control group. Participants were over 18 years old. There were randomized clinical trials (RCT) completed with original data. In addition, studies were selected without language restriction, considering works published between January 2015 and December 2022.

This review followed the PICOs question procedure to establish eligibility criteria, as shown in Table 2.

### 2.4. Study Selection

The sequence of study selection was first performed by combining MeSH terms and keywords on different databases. Subsequently, duplicate articles were rejected, and the title and abstract were read so as to identify potentially relevant articles. After reading the full paper, only the studies meeting the inclusion criteria were taken into account for this review.

Two independent reviewers were in charge of the search, article selection and data extraction process. Any discrepancies between the reviewers were discussed with a third reviewer.

The following data were extracted from each article considered for the present review: author, year of publication, characteristics of the participants, number of subjects, intervention groups, types of devices used, pathologies analyzed, outcomes, results and methodological quality.

### 2.5. Evaluation of the Quality of the Studies

The methodological quality of the studies was assessed using the PEDro scale [18] which has proved to be reliable and valid for assessing the quality of randomized controlled trials [18]. The PEDro Scale is an 11-item scale based on the Delphi list developed by Verhagen et al. [19]. One item of the PEDro Scale (eligibility criteria) was related to external validity and was not used to calculate the total score. A score equal to or higher than seven was considered high quality, a score equal to five and six was considered fair quality and a score equal to or lower than four was considered poor quality. Eventual discrepancies were discussed with a third reviewer (Table 3).

## 3. Results

### 3.1. Literature Search and Screening

Initially, the database search generated 779 articles, of which 766 studies were excluded including duplicates and those that did not meet the inclusion criteria. After reading the full text, we were left with 13 articles, of which two were also excluded. The work carried out by Lee et al. [30] was excluded because both groups received telerehabilitation treatments. The studies by Hwang et al. [22] were rejected because the variables of analysis were not of interest for the present work. The description of the selection process is shown in the PRISMA flowchart (Figure 1).

### 3.2. Quality of the Methods in the Included Studies

The quality of the methods in the included studies ranged between 5 and 8 points on the PEDro scale, from 0 to 10 (Table 1). Nine (81.81%) of the eleven included trials scored higher than 6 points on the PEDro scale. According to the PEDro scale, five studies were of fair quality [18,23,24,27,28], while six of them were of high quality [20,22,26].

Lower methodological quality scores found in the studies was due to lack of therapist blinding (*n* = 11, 100%) and lack of participant blinding (*n* = 11, 100%).

### 3.3. Characteristic of the Study

A total of 11 RCTs were included, which involved 1196 patients. The studies were grouped in eight countries from different continents; Poland [18], China [20], two in Spain [21,24], two in Australia [22,26], Canada [23], New Zealand [25], the USA [27] and two in Turkey [28,29].

Results were divided into two sections; the first section (Table 4) is organized according to participants’ sociodemographic data, the number of subjects in each intervention group and the pathology analyzed. The second section (Table 5) shows the type of device used, the time of measuring, the tests used and the results obtained.

### 3.4. Participants

The sample size ranged from 18 to 306 subjects. All studies included participants over 18 years old, with the average age being 57.35 in the TR group and 59.19 in the control group. The size of the study group that used telerehabilitation ranged between 8 [24] and 151 subjects [27]. The smallest number of participants in the control group was 10 people [24], with 153 being the highest [27].

In terms of gender, there were women participating in all studies, with a sample size of 100% in a study of breast cancer survivors [21]. Women’s smallest representation was 14.2% [25], in a study on coronary heart disease.

### 3.5. Fields of Activity

Telerehabilitation is used to treat different pathologies such as cardiac pathology [18,22,25], neurologic pathology [20,28], breast cancer [21] and musculoskeletal pathology [23,24,26,27,29].

Among cardiac pathologies, heart failure was specifically analyzed in two of the studies [18,22]. The cancer study focused on breast cancer survivors [21]. Within the neurological pathologies, exercises were prescribed for multiple sclerosis [28] and for patients with hemiplegia [20,25]. Most of the studies focused on musculoskeletal pathologies, and two of them focused on patients’ rehabilitation after knee surgery [23,27], one on hip surgery [26] and another on patients after subacromial decompression surgery [24].

### 3.6. Type of Intervention, Exercise Program

The intervention group in each study received the guideline/control of performing therapeutic exercise through telerehabilitation. All studies had a control group performing therapeutic exercise with a physical therapist.

The exercise programs for most of the studies were similar in the two groups [20,22,23,24,26], for some, the session was designed in three parts: a 5–10 min warm-up, a 10–30 min basic aerobic endurance workout, and a five minute cool-down [18,21], for others, the intervention program was based on a functional and general approach [20,21,22,23,24,25,27,28,29], and others opted for a more specific exercise program [20,21].

The time taken to perform the exercises ranged from 10 min to 30 min [18,24] and 60 min a day [21] with a frequency reported by studies showing exercise sessions of one to three per day [18,23,26,29], two times per week [20,22], three a week [18,21] and five a week [24]. The total number of scheduled sessions ranged from 8 [23] to 60 [20]. The duration of supervised exercise programs ranged from 6 weeks [26] to 12 weeks [20,22,24,25,27,28].

### 3.7. Type of Device

Most studies used videoconference systems via computer or telephone [20,22,23,28] as a direct communication strategy with the patient, including the use and support of remote telemonitoring equipment [18], which allowed for monitoring patients’ online activity and data, the storage of data [20], to have access to a messaging platform [21,29] and to email to send the exercises [24]. Others use a cloud-based virtual telehealth system that works with a three-dimensional (3D) form to observe posture and movement, using a digitally simulated trainer to demonstrate and guide activity [27]. Exercise was also delivered through more complex REMOTE-CR platforms, together with several devices, a smartphone, a portable sensor and web applications and customized middleware [25]. Phones were not used for exercise guidance but as a communication device between patients and therapists [21]. Not all telerehabilitation devices facilitate bidirectional communication when it comes to exercise execution [24,25,26,27,29], although platforms that at some point use multiple devices communicate in a bidirectional way.

### 3.8. Measures Considered

#### 3.8.1. Quality of Life

Quality of life was measured through several items [18,22,25,26,28,29]. However, the measurement instruments varied widely. Two papers used SF36 [18,29], one of them used SF12 [26] together with QOL and EQ-5D-5, three of them used EQ-5D [22,25,26] and one of them also used MLWHFQ [22].

#### 3.8.2. Physical and Functional Level: Functional Scales

The physical and functional component of the subjects were studied in 10 of the 11 items using different instruments: Barthel, MBI, BBS, BERG, Ranking Mod MRS [20] and 6MWD were included in three studies [21,22,23]. In addition, CM [24], VO2 Max [25], BOOMER [22], WOMAC [23], KOOS [23], FTST, ODI, TSK [29], PROMIS [27], FIM and NHP-I were only used in one article each. In contrast, TUG was used in several studies [22,23,26,29].

#### 3.8.3. Satisfaction

Satisfaction was measured in three of the items using three different questionnaires CSQ8 [22], SUS [26]. The health care questionnaire was measured using the HCSQ questionnaire [23].

#### 3.8.4. Adherence

Three studies were conducted to assess adherence. Data obtained from the completion of the prescribed sessions [25], the compliance and recording of the daily log-book [21] and the implementation of EARS exercise [29] were measured.

#### 3.8.5. Cost

Regarding cost, only two works studied the cost-effectiveness of the telerehabilitation intervention. One study included QALY [25], and another obtained the data by collecting data from reports of physiotherapy consultations, hospital stays or pharmacological expenses, among others [27].

#### 3.8.6. Pain

Only one article analyzed the effectiveness of telerehabilitation in pain using the VAS scale [29].

### 3.9. Limitations of the Studies Regarding the Use of Telerehabilitation

Some of the studies referred to certain limitations when using telerehabilitation. Chen et al. [20] reported that, positively, home-based telerehabilitation may be an important way of overcoming barriers and may be useful for stroke survivors living in rural areas. Although cost-effectiveness was not observed, it is likely to reduce costs and travel time. Similarly, Pastora-Bernal et al. [24] highlighted that being able to access telerehabilitation from any place and location is one of its positive aspects.

Prvu Bettger et al. [27] showed that telerehabilitation in patients undergoing knee replacement surgery had lower total costs when compared to traditional rehabilitation. Other authors, such as Maddison et al. [25], have shown the beneficial cost-effectiveness of the REMOTE-CR program in patients with coronary heart disease. In addition, reduction in drug costs through individualized interventions was observed, as well as benefits due to overcoming accessibility barriers.

Nelson et al. [26] reported that a limitation of the intervention group (telerehabilitation) was the fact that it was led by one physiotherapist, which could foster a stronger bond between the patient and the physiotherapist. However, the control group was led by several physiotherapists. Tarakci et al. [28] reported that telerehabilitation in patients with multiple sclerosis may help to improve quality of life and daily activities. Nevertheless, conventional rehabilitation was reported to be more beneficial for fatigue management.

Other authors indicated limitations related to the variables of the studies [21,22,23].

When focusing on where telerehabilitation was carried out, Hwang et al. [22] indicated that their study was conducted in a metropolitan area with good Internet connection. In addition, they reported that further research would be needed to determine the applicability of telerehabilitation in rural and remote areas with unsteady Internet connection.

Other aspects to be noted are the lack of participation in the studies [27], adherence control [26] and participants dropping out of the program [25,28].

In the same way, Piotrowitz et al. [18] reflected a negative aspect, which was the lack of follow-up of the participants in the studies. In addition, Özden et al. [29] also reflected on the lack of training for the agents involved in the program.

### 3.10. Pathologies Studied

Regarding the pathologies studied in this review, the application of physical exercise using telerehabilitation in breast cancer was only studied in one article, which indicated that it was effective in both groups. However, it was shown to be significantly effective in the TR group in terms of functional capacity and cognitive functioning, which was also maintained after 6 months.

Musculoskeletal pathology was observed in five studies: two in knees, one in hips, one in the shoulder and one in lower back pain. These works analyzed functionality, quality of life, adherence to treatment, costs and pain. When focusing on functionality, both groups improved in three of the studies [23,24,27,29]. Özden et al. [29] even achieved a significant improvement in the TR group when compared to the conventional rehabilitation group. Quality of life was measured in several ways, and a positive effect in both groups [26] and significantly in the TR group [29] were shown. Regarding satisfaction levels, in the case of patients with musculoskeletal disorders who received telerehabilitation and face-to-face rehabilitation, results were similar for both groups [23,26,29]. The studies that analyzed and compared the costs of both types of intervention for patients with this condition reflected lower overall costs for the TR group.

Cardiac pathology was observed in two studies [22,25], both focusing on functionality, quality of life and satisfaction. Maddisson et al. [25] also took treatment adherence and cost-effectiveness into account. In both studies, regarding functional level, treatment with TR was equally effective as in-person treatment in the physiotherapy room. Quality of life was satisfactory in both groups. Adherence was higher in the TR group, while costs were significantly lower in the TR group.

Neurological processes were taken into account in two studies [20,28], where functionality [20,28], quality of life, health profile and fatigue [28] were observed. The effectiveness of exercise through TR in terms of functionality was shown to be as effective as exercise delivered in a conventional way. Results regarding quality of life indicate that, as a tool, TR helps to achieve quality of life, especially in the case of patients who live far from rehabilitation centers, who were also able to reduce expenses.

### 3.11. Type of Device and Effectiveness

Most of the studies used bidirectional communication as means of intervention. Videoconferencing was used in three [23,24,26] of the five studies dealing with musculoskeletal problems, in one dealing with cardiac pathology [22], and in another one dealing with stroke survivors [20]. In some studies, the audio and video videoconferencing systems were used together with some other biofeedback and physiological data collection devices [20]. In all studies, results regarding the effectiveness of telerehabilitation versus face-to-face rehabilitation were similar in the two groups in terms of functionality, quality of life and satisfaction.

One-way devices included exercise videos uploaded using applications [21,27,29] or sent by email [24] and devices that record and send data [18]. In addition, the phone was used as a communication device [18,21]. These devices were effective in both groups in terms of functionality, quality of life, costs, and patients’ satisfaction with telerehabilitation.

## 4. Discussion

The purpose of this review was to evaluate the effectiveness of telerehabilitation as a tool for therapeutic exercise, the most common pathologies treated and the most common devices used for telerehabilitation. To this end, 11 articles were selected that met the criteria proposed for our review. It is worth highlighting the great heterogeneity in terms of the type of telerehabilitation device used, the variables measured and the pathologies studied. When focusing on the participants, the smallest sample size was 18 patients [24], while the largest was 306 [23], which is an important limitation to consider when extrapolating the results [20].

Regarding the methodological quality of the studies, we found that almost half of them obtained low scores of 5/6 points [18,23,24,27,28], highlighting that the most frequent general characteristic of all studies was the lack of blinding of participants and physiotherapists. Despite these circumstances, six of the studies are of high quality [20,21,22,25,26,29].

Therapeutic exercise was a mandatory condition for the inclusion of studies in this review, both in the intervention group performed through telerehabilitation and in the control group where it was performed in hospitals or rehabilitation centers [18,20,22], or in some cases at home without supervision, or only after receiving written guidelines [26,29].

In this review, some authors prescribed physical exercise only to recover functionality, especially when treating musculoskeletal pathologies [23,24,26,27,29]. More than half of the studies measured exercise-related quality of life through telerehabilitation [18,22,25,28]. Only two studies measured costs [25,27], while three considered participants’ satisfaction [22,23,26]. It is worth noting that only one study [29] analyzed pain when it is a variable that physiotherapists take into account as an objective in most of the treatments in different pathologies [31,32,33].

The studies reviewed [21,25,26,29] show a high degree of adherence to treatment delivered using telerehabilitation methods, as the Rutkowski et al. [34] study showed. In contrast, the study carried out by Torriani-Pasin et al. [35] found adherence to be moderate as it was negatively affected by technological problems, such as poor internet connection, or technological illiteracy, among others.

When focusing on pathologies, only four areas were studied. In the field of neurology, the studies reviewed took individualized exercises adapted to the subject’s condition [28] and the use of neuromuscular stimulation [20] into account. In addition, the prescription of individualized exercise is gaining relevance, as it has proved to have positive results, showing the effectiveness of exercise in neurology, as in the study by Halabchi et al. [36]. This indicates that a supervised, individualized exercise program can improve physical fitness, functional capacity and quality of life, as well as modifiable impairments in patients with multiple sclerosis.

It is important to highlight the different alterations caused by pathologies. Authors such as Pelliccia et al. [37] and Kampshoff et al. [38] support the effectiveness of physical exercise in patients with pathologies such as cancer. However, only the study carried out by Galiano-Castillo et al. [21] analyzed cognitive condition in which exercise was shown to be equally efficient. This study also indicated improvements at a functional level in women who completed telerehabilitation. These data are in line with the study carried out by Campbell KL et al. [39], who also looked at the effectiveness of the treatment on cognitive performance. Furthermore, as Liu et al. [40] showed in their study, the pathology itself and the side effects of the treatments mean that cancer sufferers not only have functional alterations, but also emotional changes.

Another pathology considered for the present review is the cardiac pathology, which is analyzed in three studies [18,22,25]. The novel aspect of the work carried out by Hwang et al. [22] lies in the fact that telerehabilitation is delivered in groups, an aspect that the participants value positively, favoring group interaction. As we said, all three studies show the effectiveness of exercise in cardiac pathology, focusing mainly on gait training [18], as analyzed by Batalik et al. [41], and on individualized exercises [22,25], as observed by Bei et al. [42].

Almost half of the studies refer to the treatment of musculoskeletal pathology [23,24,26,27,29]. We found that the exercise programs designed in the studies used different devices and outcomes depending on the musculoskeletal pathology. However, in general, interventions using therapeutic exercise through telerehabilitation are at least as effective as interventions using conventional physiotherapy, mainly in terms of functionality as shown in the studies on the cardiac or neurological pathology [20,22].

The studies considered for the present review analyzing the effectiveness in terms of improvement of quality of life reflected a positive change in quality of life, which is in line with what Chumbler et al. [43] have reported regarding an improvement in quality of life in patients who used telerehabilitation.

Many digital devices are available for the implementation of telerehabilitation programs, but videoconferencing systems [20,21,22,23,28] are commonly used, sometimes accompanied by monitoring and data recording systems [18,20]. Lawford et al. [44] reported on the advantages of videoconferencing over other devices, with both patients and physiotherapists describing mostly positive experiences using a videoconferencing-based Internet. Nevertheless, as Huang et al. [45] have pointed out, the choice of treatment should reflect preferences, anticipation, risk profile, funding and ease of accessibility to health services.

According to the results reported in the work carried out by Suso-Martí et al. [46], in order to favor adherence and the effectiveness of exercise programs, a paradigm shift is needed on the part of the physiotherapist, which includes, among other things, therapeutic skills, method and device training and good communication with the patient.

The advantages of lower cost and less interference of rehabilitation processes in patients’ daily lives, according to the report of Stefanakis et al. [47], indicate that the risk of adverse effects seems very low and could justify the implementation of telerehabilitation in controlled clinical settings [43]. However, like Batalik L et al. [41], we think that addressing telerehabilitation clinical safety requires more emphasis and further research.

### Limitations of the Study

This systematic review has certain limitations. One of them is that some relevant articles may have not been considered as a result of not searching on other databases. It was also difficult to extrapolate the results obtained for each pathology to the population, since not many of the works considered offering exercise treatments using telerehabilitation. In addition, the fact that the diseases within each pathological condition are different makes it difficult to compare results. Other methodological limitations, such as the small sample size or the lack of a follow-up registry, could affect the results of this review. Future studies should describe and standardize the measurement tools for each pathology so as to facilitate comparison and replicability. Furthermore, effectiveness should not only be observed in terms of functionality and quality of life, but also in terms of satisfaction and cost/effectiveness.

## 5. Conclusions

Most of the studies concluded that physiotherapy interventions using telerehabilitation were at least as effective as traditional rehabilitation interventions and are considered feasible and effective options. In general, telerehabilitation interventions were shown to improve functional level and quality of life, as no significant differences with control groups were observed. This review provides information on patients’ high levels of satisfaction and adherence, with values equivalent to traditional rehabilitation in all cases. However, such variable was not considered in all cases. Another conclusion drawn from our work is that there are few studies in which telerehabilitation is used to apply therapeutic exercise. This shows that we have not really learned from the time of the pandemic, when different telerehabilitation tools were generally implemented in the treatment process. The results of this study show that prescribed telerehabilitation exercise is as effective as face-to-face exercise, and can set a basis for possible telerehabilitation practices in a post-pandemic world.

## Figures and Tables

**Figure 1 ijerph-20-04358-f001:**
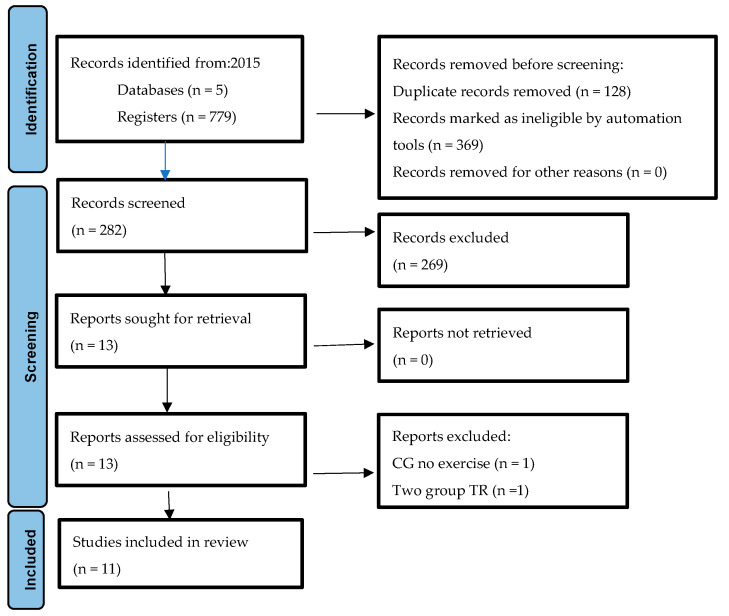
Flowchart diagram of the study.

**Table 1 ijerph-20-04358-t001:** Research strategy in databases.

Database	Strategy	Results
Pubmed	(“telerehabilitation”[MeSH Terms] OR “telerehabilitation”) AND (“Exercise Therapy”[Mesh] OR “Exercise Therapy”)	295
Embase	(“telerehabilitation”/exp OR telerehabilitation) AND “exercise therapy”: ti,ab,kw	51
PEDro	telerehabilitation exercise	88
Scopus	(TITLE-ABS-KEY (telerehabilitation) AND TITLE-ABS-KEY (“exercise therapy”)	338
SPORTDiscus	TI telerehabilitation AND AB exercise therapy	7

**Table 2 ijerph-20-04358-t002:** Details and PICOs composition for the inclusion of studies.

Parameters	Description
Review Question	Therapeutic physical exercise programs carried out through telerehabilitation are as effective as face-to-face exercise.
Population	The people in this study are over 18 years of age and have been prescribed an exercise program as a treatment.
Intervention	Supervised physical exercise program through telerehabilitation
Comparator	Physical exercise program carried out on-site
Outcomes	Functionality, quality of life, level of satisfaction, adherence to treatment, pain and costs.
Study Design	Randomized clinical trials (RCT)

**Table 3 ijerph-20-04358-t003:** Methodological quality PEDro.

Trial	1	2	3	4	5	6	7	8	9	10	PEDro Score
Piotrowicz E. 2015 [18]	Yes	No	Yes	No	No	No	Yes	No	Yes	Yes	5
Chen J. 2017 [20]	Yes	Yes	Yes	No	No	Yes	Yes	Yes	Yes	Yes	8
Galiano-Castillo N. 2017 [21]	Yes	Yes	Yes	No	No	Yes	Yes	Yes	Yes	Yes	8
Hwuang R. 2017 [22]	Yes	Yes	Yes	No	No	Yes	Yes	Yes	Yes	Yes	8
Moffet H. 2017 [23]	Yes	No	Yes	No	No	Yes	Yes	No	Yes	Yes	6
Pastora-Bernal JM. 2018 [24]	Yes	No	Yes	No	No	Yes	Yes	No	Yes	Yes	6
Maddison R. 2019 [25]	Yes	Yes	Yes	No	No	Yes	Yes	Yes	Yes	Yes	8
Nelson M. 2020 [26]	Yes	No	Yes	No	No	Yes	Yes	Yes	Yes	Yes	7
Prvu Bettger J. 2020 [27]	Yes	No	Yes	No	No	No	Yes	Yes	Yes	Yes	6
Tarakci E. 2021 [28]	Yes	No	Yes	No	No	Yes	No	No	Yes	Yes	5
Özden F. 2022 [29]	No	Yes	Yes	No	No	Yes	Yes	Yes	Yes	Yes	8

1—Subjects were randomly allocated to groups (in a crossover study, subjects were randomly allocated an order in which treatments were received); 2—Allocation was concealed; 3—The groups were similar at baseline regarding the most important prognostic indicator; 4—There was blinding of all subjects; 5—There was blinding of all therapists who administered the therapy; 6—There was blinding of all assessors who measured at least one key outcome; 7—Measures of at least one key outcome were obtained from more than 85% of the subjects initially allocated to groups; 8—All subjects for whom outcome measures were available received the treatment or control condition as allocated, or, where this was not the case, data for at least one key outcome were analyzed by “intention to treat”; 9—The results of between-group statistical comparisons were reported for at least one key outcome; 10—The study provides both point measures and measures of variability for at least one key outcome.

**Table 4 ijerph-20-04358-t004:** Detailed information of selected studies.

Author and Year	Country	Participants	N Gender %	Age (Average)	Sample Size Groups	Study Objectives	Pathology	Quality
Piotrowicz E. 2015 [18]	Warsaw (Poland)	152	M: 117 (65) TR 64 CG 53 F: 63 (35) TR 21 CG 42	TR 56.4 ± 10.9 CG 60.05 ± 8.8	TR: 77 GC: 75	To assess changes in the quality of life of patients with heart failure	Cardiac	5
Chen J. 2017 [20]	Shangahi (China)	54	M: 33 (61.11) TR 18 CG 15 F: 21 (38.89) TR 9 CG 12	TR 66.52 ± 12.08 CG 66.15 ± 12.33	TR: 27 CG: 27	To assess physical function and determine whether it can be helpful to caregivers	Neurologic	8
Galiano-Castillo N. 2017 [21]	Granada (Spain)	81	F: 76 (100) TR 39 CG 37	TR 47.4 ± 9.6 GC 49.2 ± 7.9	TR: 40 CG: 41	To improve functional capacity and cognition	Breast cancer	8
Hwang R. 2017 [22]	Brisbane (Australia)	53	M: 40 (75) TR 19 CG 21 F: 13 (25) TR 5 CG 8	TR 68 ± 14 CG 67 ± 11	TR: 24 CG: 29	To prove non-inferiority in terms of functional capacity, muscle strength, quality of life, patient satisfaction and attendance rates	Cardiac	8
Moffet H. 2017 [23]	Québec (Canada)	205	M: 89 (48.9) TR 35 CG 54 F: 93 (51.1) TR 49 CG 44	TR 65 ± 8 CG 67 ± 8	TR: 84 CG: 98	To compare patients’ satisfaction levels	Musculoskeletal; total knee arthroplasty	6
Pastora-Bernal JM. 2018 [24]	Málaga (Spain)	18	M: 10 (55.55) TR 4 CG 6 F: 8 (44.45) TR 4 GC 4	TR 49.63 ± 10.08 CG 54.8 ± 11.84	TR: 8 CG: 10	To assess the feasibility and effectiveness of customizable telerehabilitation intervention	Musculoskeletal (shoulder)	6
Maddison R. 2019 [25]	Auckland y Tauranga (New Zealand)	162	M: 139 TR 69 CG 70 F: 23 (14.2) TR 13 CG 10	TR 61.0 ± 13.3 CG 61.5 ± 12.2	TR: 82 CG: 80	To compare the effects and costs of cardiac telerehabilitation	Cardiac	8
Nelson M. 2020 [26]	Brisbane (Australia)	70	M: 26 (37.14) TR 12 CG 14 F: 44 (62.86) TR 23 GC 21	TR 62 ± 9 CG 67 ± 11	TR: 35 CG: 35	To determine whether outpatient physiotherapy using telerehabilitation is as effective as face-to-face physiotherapy after total hip replacement	Musculoskeletal PTC	7
Prvu Bettger J. 2020 [27]	North Caroline (USA)	306	M: 114 (37.5) TR 61 CG 53 F: 190 (62.5) TR 90 GC 100	TR 65.4 ± 7.7 CG 65.1 ± 9.2	TR: 151 CG: 153	To determine whether outpatient physiotherapy using telerehabilitation is as effective as face-to-face physiotherapy	Musculoskeletal; total knee arthroplasty.	6
Tarakci E. 2021 [28]	Estambúl (Turkey)	41	M: 7 (23.33) TR 4 CG 3 F: 23 (76.66) TR 11 GC 12	TR 39.46 ± 10.59 CG 41.00 ± 11.09	TR: 15 CG: 15	To evaluate the effectiveness of telerehabilitation on fatigue, health status, quality of life and daily life activities	Neurologic; multiple sclerosis	5
Özden F. 2022 [29]	Muğla Sıtkı Koçman (Turkey)	54	M: 19 (38) TR 11 CG 8 F: 31 (62) TR 14 GC 17	TR 40.1 ± 1.6 CG 42.3 ± 1.6	TR: 25 CG: 25	To assess pain, function, quality of life, expectations, satisfaction and motivation in patients with chronic low back pain	Musculoskeletal; chronic low back pain	8

TR: Telerehabilitation; CG: Control group; M: Male; F: Female.

**Table 5 ijerph-20-04358-t005:** Characteristics of telerehabilitation devices, time of measuring, outcome and result.

Author and Year	Type of Device	Type of Exercise	Outcome	Time of Measuring	Session Frequency/Treatment Duration	Test	Result
Piotrowicz E. 2015 [18]	Remote equipment for telemonitoring and supervised exercise training (EHO 6 device transmit the ECG) and a mobile phone	Cardiac rehabilitation through gait training. Telematic TR and CG with cycloergometer.	Quality of life	At baseline and at 8 weeks, Quality of life	3 times per week/8 weeks	SF-36	Both groups significantly improved quality of life. GI patients improved mainly in mental categories. GC improved their overall physical well-being.
Chen J. 2017 [20]	Audio-video system (videoconferencing), biofeedback instrument and data logging	TR individualized physical exercise plan + neuromuscular stimulation (ETNS) and CG performs ambulatory RHB with the same type of exercises as TR.	Functionality	At baseline, at end (12 weeks) and follow-up at 24 weeks	1 h 2 times per day/12 weeks	BARTHEL, MBI, BBS, MRS	TR as effective as conventional RHB for functional recovery in stroke. It is a way to overcome barriers for stroke survivors living in rural areas. It is likely to reduce costs and travel times.
Galiano-Castillo N. 2017 [21]	Web application “e-CUIDATE”	Both groups exercise program aimed at functional and cognitive recovery	Functionality, Adherence to treatment	At baseline, at the end of the intervention (8 weeks) and follow-up (6-month follow up)	90 min per day, 3 times per week/8 weeks	6MWT, ACT, TMT, Diarios de ejercicios	Both groups showed improvements in 6MWT and also differences between groups with better results for TR. ACT total TR improves compared to CG. TMT unchanged.
Hwang R. 2017 [22]	Program web-based exercises using videoconferencing software	TR cardiac RHB exercises and education in real time and CG traditional hospital program with same frequency and same duration as TR	Functionality, quality of life, level of satisfaction	At start-up, 12 weeks and 24 weeks	Twice a week/12 weeks	6MWD, BOOMER, TUGT, EQ-5D, MLWHFQ, CSQ-8	The intervention is at least as effective as rehabilitation without telerehabilitation, promotes higher frequency of attendance and improves equity of access to cardiac RHB programs
Moffet H. 2017 [23]	TR platform and videoconferencing system	Similar exercise program for both groups based on functionality. Exercises aimed at mobility recovery and strengthening.	Functionality, level of satisfaction	Before surgery, in the hospital, after physical therapy and at 4 months.	16 interventions of 45 to 60 min every 2 weeks, for 8 weeks/8 weeks	HCSQ, WOMAC, KOOS, 6MWD,	Similar level of satisfaction between IG and CG. The greater the improvements in WOMAC and KOOS, the higher the level of satisfaction. The use of TR improves access to rehabilitation services.
Pastora-Bernal JM. 2018 [24]	Web and videoconferencing-based system	Both groups strengthening and joint amplitude exercises, web-based TR and regular face-to-face physiotherapy CG	Functionality	At baseline, 4, 8 and 12 weeks	5 days per week/12 weeks	Constant-Murley Test	Evidence of efficacy of TR, physical and functional improvements in both groups. Non-significant trend of greater improvements in TR.
Maddison R. 2019 [25]	REMOTE-CR telerehabilitation system	TR individualized exercise using customized platform and traditional cardiac CG RHB	Functionality, Quality of life, Adherence to treatment, Cost-utility analysis	Start-up, 12 weeks and 24 weeks	12 weeks	VO2 max, EQ-5D, Exercise adherence was calculated as the completion of prescribed exercise sessions prescribed. QALY	Efficient and cost-effective alternative, individualized intervention and overcoming accessibility barriers, implementation costs of the REMOTE-CR program were substantially lower than those of CBexCR.
Nelson M. 2020 [26]	Apple iPad technology and Wellpepper clinic internet application and eHAB (real time videoconferencing)	CG strengthening and gait re-education exercises. TR same exercise content as GC via internet application.	Quality of life, Functionality, Level of satisfaction	Onset (before surgery), at discharge from hospital, 6 weeks and 6 months after operation	3 times per week/6 weeks	QOLS, HOOS, TUG, SF-12, EQ-5D-5L, satisfaction survey, SUS	Easy access for the population, high levels of satisfaction, physical and functional results not inferior to those obtained with traditional physical therapy. Program compliance rate in favor of TR
Prvu Bettger J. 2020 [27]	VERA (virtual exercise rehabilitation assistant)	Prescribed exercises varied by therapist and patient in both groups. Sports and recreational exercises (squatting, running, jumping, twisting/pivoting, kneeling) were included.	Functionality, cost	At baseline, 6 weeks, 12 weeks	Frequency and duration NOT restricted/12 weeks	KOOS, PROMIS, Cost analysis.	Lower costs in TR, fewer rehospitalizations than in GC, no inferiority in terms of knee flexion–extension and gait speed.
Tarakci E. 2021 [28]	Video calls	Functionality, quality of life, health profile, fatigue	Functionality, quality of life, health profile, fatigue	At baseline and end (12 weeks)	3 sessions per week/12 weeks	FIM, NHP-I, FSS, QOLS	Telerehabilitation can help improve quality of life and activities of daily living although supervised exercise without TR may be more beneficial for fatigue and health profile.
Özden F. 2022 [29]	Exercise videos Fizyoweb system	Both groups: lumbar and lower back stretching exercises, abdominal strengthening, spinal column mobility.	Functionality, Quality of life, Adherence to treatment, Pain	At baseline and 8 weeks	once daily for 8 weeks	TUG, FTST, ODI, TSK, SF-36, EARS, VAS	The TR protocol has a positive effect on all clinical parameters (pain, functionality, quality of life, kinesiophobia, motivation, satisfaction) compared to conventional RHB.

SF-36, Medical Outcome Survey Short Form 36; MBI, Barthel Modified; BBS, Berg Balance Scale; MRS, global disability; CSI, Caregiver Strain Index; RMS, value of target; BOOMER, balance outcome measure for elder rehabilitation; EQ-5D, EuroQoL; MLWHFQ, Minnesota Living With Heart Failure questionnaire; RUIS, Revised Urinary Incontinence Scale; TUGT, Timed Up and Go test; VAS, visual analogue scale; 6MWT, Six Minute Walking Test; 6MWD, 6-min walk distance; MLWHFQ, Minnesota Living with Heart failure Questionnaire; CSQ-8, Client Satisfaction Questionnaire; HCSQ SF-36, Medical Outcome Survey Short Form 36; RMS, value of target; ACT, Auditory Consonant Trigrams; QALY, Quality-Adjusted Life Year; TMT, TrailMaking Test; HCSQ, Health Care Satisfaction Questionnaire; WOMAC, Western Ontario and McMaster Universities Osteoarthritis Index; KOOS, The Knee injury and Osteoarthritis Outcome Score; SF-12, Short Form-12; EQ-5D-5L, Quality of live; SUS, usability scale; CSQ-8 Client Satisfaction Questionnaire; PROMIS, the Patient-Reported Outcomes Measurement Information System; FIM, functional independence measure; NHP-I, first section of Nottingham Health Profile; FSS, fatigue severity scale; QOLS, quality of life scale; QALY, quality-adjusted life years; FTST, Five Times Sit to Stand; ODI, Oswestry Disability; TSK, kinesiophobia score; EARS, Exercise Adherence Rating Scale; THR, Introduction Total hip replacement.

## Data Availability

Not applicable.

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
