# Peer review of "Telerehabilitation as a Therapeutic Exercise Tool versus Face-to-Face Physiotherapy: A Systematic Review"

_ijerph, 2023, doi:10.3390/ijerph20054358_

Round 1

Reviewer 1 Report

This is a systematic review that aims to evaluate the effectiveness of telerehabilitation (AKA digital physiotherapy/ physiotherapy delivered through telecommunication methods).

Title: Clear title.

Abstract: Abstract is clear, but the methods section is too long. May help to condense methods section and further add more detail results section.

Introduction: Good rationale established given the context of the pandemic. However, it may be helpful to elaborate on more research gaps that exist with research in telerehabilitation. What will this systematic review accomplish?

Methods:  Methods section is well written.

Results: Section 3.1 of the results section has many grammatical errors—it is a run-on sentence.

In Table 4, please explain the abbreviated terms (eg. TR, CG, GC). It may also be helpful to add a row in Table 4 that describes the brief aim of each included study, so the reader is able to follow.

There are a few spacing errors in section 3.4 and 3.5, as well as in most of section 3.8.

This section may benefit from more detail into the themes studied. It may also be helpful to add a section that discusses potential drawbacks of telerehabilitation that were mentioned in the included studies. Please review this section accordingly.

Discussion: May be helpful to summarize the results of the study more briefly in the first part of the discussion, as it is too long. What are the main takeaways? Limitations need more elaboration, and the results should be compared to other previous studies/reviews. It may also be really insightful if the results of this study could inform potential telerehabilitation practices in a post-pandemic world. This section may need more work and revision.

Conclusion: Needs more work, and it may be helpful to discuss the study implications on a larger scale.

Author Response

Thank you for all comments that will improve the quality and understanding of the review. We will respond to every comment from reviewer 1.

Comment 1. This is a systematic review that aims to evaluate the effectiveness of telerehabilitation (AKA digital physiotherapy/ physiotherapy delivered through telecommunication methods).

Title: Clear title.

Response 1. Thank you.

Comment 2. Abstract: Abstract is clear, but the methods section is too long. May help to condense methods section and further add more detail results section.

Response 2. Thank you for this comment, it has been modified based on your suggestions.

Comment 3. Introduction: Good rationale established given the context of the pandemic. However, it may be helpful to elaborate on more research gaps that exist with research in telerehabilitation. What will this systematic review accomplish?

Response 3: Thank you for your input, we expanded and adapted the introduction based on your comments.

Comment 4. Methods:  Methods section is well written.

Response 4. Thank you.

Comment 5. Results: Section 3.1 of the results section has many grammatical errors—it is a run-on sentence.

Response 5. Thank you for your input. It has been rewritten to make it easier to understand. Thank you

Comment 6. In Table 4, please explain the abbreviated terms (eg. TR, CG, GC). It may also be helpful to add a row in Table 4 that describes the brief aim of each included study, so the reader is able to follow.

Response 6.  The explanation of the acronyms has been included and the objectives of each study have also been included in the table. Thank you for your suggestions.

Comment 7. There are a few spacing errors in section 3.4 and 3.5, as well as in most of section 3.8.

Response 7. We hope we have solved the problem with the layout, it is constantly getting out of order.

Comment. 8. This section may benefit from more detail into the themes studied. It may also be helpful to add a section that discusses potential drawbacks of telerehabilitation that were mentioned in the included studies. Please review this section accordingly.

Response 8. In line with your feedback, we have included a column in table 4 that includes the objectives of each article included in the review. In addition, we have added a section on the analysis of the drawbacks reflected in the studies included in this review in relation to telerehabilitation.

Comment 9. Discussion: May be helpful to summarize the results of the study more briefly in the first part of the discussion, as it is too long. What are the main takeaways? Limitations need more elaboration, and the results should be compared to other previous studies/reviews. It may also be really insightful if the results of this study could inform potential telerehabilitation practices in a post-pandemic world. This section may need more work and revision.

Response 9. Thank you for your input, we have redrafted the discussion and limitations based on your suggestions.

Comment 10. Conclusion: Needs more work, and it may be helpful to discuss the study implications on a larger scale.

Response 10. We are redrafting the conclusions and including your input.

Thank you for giving us the possibility to improve the article for publication.

Reviewer 2 Report

The study is well structured and the objective is aligned with the article.

- Improve abstract conclusion;

- Include the description of all acronyms;

- Did you standardize the inclusion method of studies for rehabilitation time?

- It would be interesting to focus the study on specific pathologies, for example, only studies with cardiorespiratory diseases.

Author Response

Thank you for all comments that will improve the quality and understanding of the review. We will respond to every comment from reviewer 2.

  • Comment 1. The study is well structured and the objective is aligned with the article.

          Response 1. Thank you.

  • Comment 2. Improve abstract conclusión

          Response 2. The conclusions section of the summary has been redrafted.

  • Comment 3. Include the description of all acronyms

          Response 3. We have included the description of the acronyms.

  • Comment 4. Did you standardize the inclusion method of studies for rehabilitation time? Response 4. Rehabilitation time has not been taken into account in the inclusion of studies.
  • Comment 5. It would be interesting to focus the study on specific pathologies, for example, only studies with cardiorespiratory diseases.

           Response  5. You are right, although in this review it has been raised to resolve the question of whether the exercises prescribed by telerehabilitation versus face-to-face are effective, to see in which pathologies they are applied and the type of instrument used. It is a good idea as a future line of research now that we have the results of this review.

Thank you for giving us the possibility to improve the article for publication.

Reviewer 3 Report

In the title, it is important to note that the comparison is against a face-to-face exercise, not against a non-exercise. Please adjust the t

The introduction is a presentation of the evolution of telerehabilitation and then jumps abruptly to the objective of the review. Normally the introduction is structured as follows: presentation of the current situation, problem/situation in the literature that is not solved and, finally, how our article will solve this problem. Please restructure the introduction to make it more readable.

The present study is almost completely unrelated to the Prospero record. Please review this topic

Selecting the time range and search terms does not yield the same number of results. In addition, it is not understood and/or explained why only articles from 2015 are used.

Please redo figure 1 with straight lines.

Section 3.8 of Results explains what is measured but not the positive or negative results of the selected studies.

The discussion section focuses a lot on talking about the types of exercises and not on their effectiveness, which is the main objective of the paper.

Author Response

Thank you for all comments that will improve the quality and understanding of the review. We will respond to every comment from reviewer 3.

  • Comment 1. 1the title, it is important to note that the comparison is against a face-to-face exercise, not against a non-exercise.

Response 1. Thank you, we have modified the title based on your suggestions.

  • Comment 2. The introduction is a presentation of the evolution of telerehabilitation and then jumps abruptly to the objective of the review. Normally the introduction is structured as follows: presentation of the current situation, problem/situation in the literature that is not solved and, finally, how our article will solve this problem. Please restructure the introduction to make it more readable.

Response 2. Thanks for your input, the introduction has been restructured to include your suggestions.

  • Comment 3.The present study is almost completely unrelated to the Prospero record. Please review this topic

Response 3. You are right, this topic is being reviewed in prospero. https://www.crd.york.ac.uk/prospero/display_record.php?ID=CRD42021241394

  • Comment 4. Selecting the time range and search terms does not yield the same number of results. In addition, it is not understood and/or explained why only articles from 2015 are used.

Response 4. The selection of the time range and search terms have been collated again and reflected in the manuscript. The time range since 2015 is because there is a similar review to the one we conducted.  https://doi.org/10.1177/1357633X15572201

  • Comment 5.Please redo figure 1 with straight lines.

Response 5 Straight lines have been included in Figure 1.

  • Comment 6. Section 3.8 of Results explains what is measured but not the positive or negative results of the selected studies.

Response 6. In order to respond to your suggestions, a further section has been included in the results.

  • Comment 7. The discussion section focuses a lot on talking about the types of exercises and not on their effectiveness, which is the main objective of the paper.

Response 7. Thanks for the suggestions, we have modified the discussion to address your comments.

Thank you for giving us the possibility to improve the article for publication.

Round 2

Reviewer 1 Report

Thank you very much for your revisions. My comments have been addressed. 

Reviewer 3 Report

Following the comments made, the article has improved in structure and is now suitable for publication